# How socioeconomically disadvantaged people access, understand, appraise, and apply health information: A qualitative study exploring health literacy skills

Coraline Stormacq [1,2]* , Annie Oulevey Bachmann[2], Stephan Van den Broucke[3‡], Patrick Bodenmann[1‡]

1 University Center for General Medicine and Public Health (Unisanté), Department of Vulnerabilities and Social Medicine, University of Lausanne, Lausanne, Switzerland, 2 La Source School of Nursing, HES-SO, University of Applied Sciences and Arts, Western Switzerland, Lausanne, Switzerland, 3 Faculty of Psychology and Educational Sciences, Psychological Sciences Research Institute (IPSY), Catholic University of Louvain, Louvain-la-Neuve, Belgium

☯ These authors contributed equally to this work.
‡ SVD and PD also contributed equally to this work.
* c.stormacq@ecolelasource.ch

**Data Availability Statement:** Interviews transcripts in French were deposited on the SWISSUbase repository (URL: https://swissubase.ch). The

# Abstract

## Objectives

Health literacy, or a person's competence to access, understand, appraise and apply health information, can be considered a mediating factor between socioeconomic characteristics and health disparities. Socioeconomically disadvantaged people in particular present with less health literacy skills. To develop targeted interventions tailored to their real needs, it is important to understand how they function and what difficulties they encounter when dealing with health information. The purpose of this study was to explore their experiences when accessing, understanding, appraising, and applying health information in their everyday lives.

## Methods

Semi-structured face-to-face interviews were conducted with 12 socioeconomically disadvantaged adults living in the community in Switzerland (age range: 44–60 years old).

## Results

Thematic analysis of the interviews yielded four themes, describing the health literacy processes of participants, related barriers, and compensatory strategies used: Financial insecurity triggers the need for health information; Pathway 1: Physicians as ideal (but expensive) interlocutors; Pathway 2: The internet as a suboptimal alternative; and Pathway 3: Relatives as a default resource. The progression of socioeconomically disadvantaged people in the health literacy process is like an 'obstacle course', with numerous steps taken

project titled 'Stratégies utilisées par les personnes socioéconomiquement défavorisées afin d'accéder, comprendre, évaluer et appliquer les informations de santé: une étude qualitative descriptive thématique' is registered under number 'Study 20358'. The data set (number 2270) are available at the following doi: https://doi.org/10.48657/1eqm-cm37.

**Funding:** The authors received no specific funding for this work.

**Competing interests:** The authors have declared that no competing interests exist.

backwards before they can develop compensatory strategies to overcome the barriers to obtaining health information.

## Conclusions

Financial deprivation seems to be the most important factor contributing to health literacy barriers. Appraising health information is the health literacy skill with which socioeconomically disadvantaged people struggle the most. Physician-based, individual skills-based, organizational, and policy-based interventions are needed to help them overcome their health literacy challenges.

## Introduction

Health literacy (HL) is an increasingly significant topic, discussed worldwide as part of current health, social and educational policies [1], and one of the most important in healthcare and public health [2, 3]. HL is defined as '*people's knowledge, motivation and competences to access, understand, appraise, and apply health information in order to make judgments and take decisions in everyday life concerning healthcare, disease prevention and health promotion to maintain or improve quality of life during the life course*' [3].

The Integrated Model of Health Literacy [3] describes the pathways linking HL to health behaviors and outcomes as well as the determinants of and factors influencing HL levels (the antecedents) and their associated health outcomes (the consequences). This theoretical framework, predominantly used in Europe and particularly in Switzerland, also describes the four core competences of HL as follows: '*Access refers to the ability to seek, find and obtain health information; Understand refers to the ability to comprehend the health information that is accessed; Appraise describes the ability to interpret, filter, judge and evaluate the health information that has been accessed; and Apply refers to the ability to communicate and use the information to make a decision to maintain and improve health*' [3]. These four competences can be activated across the three domains of the health continuum: healthcare, disease prevention, and health promotion. The strength of this model is that it considers HL not only from an individual or clinical perspective, but also from a public health and life-course perspective. In addition, it considers HL as a process. Progressing through the four competences starts by empowering individuals and populations to be more autonomous, taking control of their health and health-related decisions [4], and continues by enabling them to engage in health promotion interventions. As such, HL is viewed as an asset, enabling more active participation in society and greater control over everyday events [4].

Research has revealed that nearly half of the adult US population has low HL levels [5]. In Europe, the recent European Health Literacy Population Survey 2019–2021 (HLS$_{19}$) showed that 13% of the population has inadequate HL skills, and 33% problematic HL skills [6]. The Health Literacy Survey Switzerland conducted in 2019–2020 (HLS$_{19-21}$-CH) showed that 38% of the population had 'problematic' HL levels, and 11% having an insufficient HL level [7]. HL has thus been referred to as a 'silent epidemic' [3, 8].

Many studies have shown that HL levels are an important predictor of health behaviors, health-related outcomes, and health status [9–11]. HL is therefore recognized as a key health determinant [12]. Poor HL levels have been associated with a range of poor health-related outcomes, such as poorer self-reported health status, poorer mental health status, higher rates of chronic disease and poorer self-management skills, higher rates of adverse health behaviors, increased mortality, and higher healthcare costs [6, 13, 14].

The Health Literacy Survey Europe (HLS-EU) report also highlighted that social and socioeconomic factors contribute to low HL levels, revealing a greater proportion of people with limited HL skills among disadvantaged social groups than in the rest of the population. Indeed, limited HL skills are influenced by a range of unfavorable social and socioeconomic characteristics, such as a low educational level, low income, poor employment status, belonging to an ethnic minority, living in a rural area, or low perceived social status [15]. People with a disadvantaged socioeconomic status are therefore particularly vulnerable to low HL, pointing to a social gradient in HL levels [15]. Particularly in Switzerland, the $HLS_{19\text{-}21}$-CH survey showed that low HL levels are closely linked to a lack of financial resources [7]. This is of concern, given the way the Swiss health system is financed. Switzerland is the third best performing health system in the world [16]. Despite the availability of universal health insurance coverage [17], its health system is paradoxically also one of the most expensive [18], and socioeconomic disparities remain one of the main causes of unequal in access to care [19]. As health insurance is compulsory and premiums must be paid independently of income, people with lower incomes pay disproportionately more for health insurance than people with higher incomes [17]. While health insurance covers the costs of medical treatments and hospitalizations, a part of the treatment costs are borne by patients themselves through a flat, annual deductible fee which amounts to a co-payment of 10% of all bills (up to a maximum of CHF 700 per year) [20].

A recent integrative review [21] showed that social and socioeconomic disadvantage were associated with poor HL levels and that HL partially mediates the relationships between the factors of socioeconomic disadvantage and disparities in health-related behaviors, health-related outcomes, and access to and use of healthcare. HL can be seen as '*leverage for action on the social determinants contributing to health inequalities and disparities*' [21]. HL thus appears to be a promising strategy for achieving greater health equity.

Yet despite the growing recognition of HL, especially so during the COVID-19 pandemic, the way in which socioeconomically disadvantaged people access, understand, appraise, and apply health information is still poorly understood. Moreover, little is known about the development of effective HL interventions among vulnerable populations like the socioeconomically disadvantaged. One systematic review aiming to assess the effectiveness of HL interventions in Europe pointed out that the most effective interventions and their related outcomes (e.g., knowledge, skills, self-efficacy, motivation, health status, health behaviors, costs) had yet to be properly identified [1]. Systematic reviews examining the effectiveness of HL interventions on health outcomes, specifically among the socioeconomically disadvantaged, are scarce. Existing reviews offer some insights, but they suffer from a lack of conclusive results [22], or address HL in a functional way (reading, writing, and using numbers effectively) and not as a broader concept [14, 23, 24]. Specifically focused on socioeconomically disadvantaged people, one recent systematic review identified the components of HL interventions associated with improved health-related outcomes [25]. This review showed that HL interventions are more likely to be effective if they are theory-based, use a person-centered approach, and combine five essential operational components (cultural appropriateness, tailoring, skills training, goal setting and active discussions) [25]. However, due to a lack of evidence, the conclusions of this review were weak. Developing effective interventions to improve HL among socioeconomically disadvantaged people requires therefore an understanding of how they function and what difficulties they encounter with health information.

To explore HL skills among socioeconomically disadvantaged people in more detail and to understand the HL-related difficulties or barriers they encounter, qualitative methods may be the most appropriate approach. Existing qualitative studies have provided some insights into this matter but have not yet provided a sufficiently detailed understanding of the mechanisms

leading to low HL levels among socioeconomically disadvantaged people [26–29]. This may be because thus far most qualitative studies have been limited to HL's *Access* competence, or only addressed HL from a functional perspective, while not considering other HL skills. In addition, and to our knowledge, there are no studies that explore HL among socioeconomically disadvantaged individuals as a comprehensive concept as proposed by the conceptual model of the HLS-EU [3]. Two studies explored health information-seeking behaviors among ethnic minorities [26, 27]. The results showed that individuals accessed health information through a multitude of channels. However, health professionals, followed by family members or friends, remained the most preferred sources of health information. Language barriers, use of medical jargon, and lack of time during medical consultations were identified as the main barriers to accessing and understanding health information. Another qualitative study among low-income users of a community health center and at risk for cardiovascular disease showed similar results [28]. Finally, a study investigated how socioeconomically disadvantaged people understand and process information related to cardiovascular health risks delivered through an interactive website [29]. The results showed that numerical information about cardiovascular risk factors was poorly understood and often underestimated, and that respondents lacked the medical knowledge to fully understand the health risks. Although this study aimed to explore health information processing, critical appraisal and decision-making skills, they could not be described.

A qualitative approach to exploring the HL processes of socioeconomically disadvantaged people can help to understand why these people are particularly vulnerable to low HL levels and how they overcome HL related challenges. This is a necessary step towards developing targeted interventions tailored to their real needs, to both improve HL skills and reduce health disparities. The present study aimed to explore the experiences of socioeconomically disadvantaged people when accessing, understanding, appraising, and applying health information in their everyday lives and the barriers they encountered when dealing with health information.

## Methods

To address the above issues, a descriptive qualitative design was used [30], involving thematic analysis [31]. The study included socioeconomically disadvantaged adults living in the community in Switzerland's French-speaking cantons of Vaud and Fribourg. As socioeconomic status is a multidimensional concept, its three main contributing factors—education, employment, and income–were used to define low socioeconomic status in the Swiss context. Inclusion criteria were: (a) not having completed compulsory schooling or having completed compulsory schooling but no post-compulsory education [32]; and/or (b) being unemployed or in a precarious employment situation (fixed-term contract, temporary job, part-time or on-call work, employment with a variable income, pseudo-independently employed) [33]; and/or (c) being in a precarious economic situation and having an income below the minimum subsistence level (inferior to a gross monthly income of CHF 2,239) [34], or receiving unemployment or social integration benefits (social assistance benefits for people in great financial difficulty who cannot meet their basic needs) [35]; (d) having sufficient French language skills to complete the study; and (e) being able to provide inform consent.

Purposive sampling was used to recruit participants with a range of different disadvantaged socioeconomic statuses and to obtain maximum heterogeneity. Recruitment occurred in the French-speaking part of Switzerland, and in three different settings to provide a diversity of situations and experiences on the phenomenon of interest: (a) people attending courses at the '*Reading and Writing Association*' [Association Lire et Écrire] because they have a low educational level or difficulties with reading, writing, or numeracy (low educational level); (b)

individuals assisted by an unemployment agency mainly dedicated to people in precarious employment situations or job seekers, and offering specialized labor market services and job placement (precarious employment status); and (c) people dependent on social security and supported by cantonal social services, which deliver social benefits to people facing financial difficulties and urgent social assistance to people in precarious situations (low income). The recruitment involved a variety of methods. Beneficiaries of the unemployment agency or social services who met the inclusion criteria received a letter outlining the study's purpose and providing the principal investigator's name and contact information. To recruit participants from the '*Reading and Writing Association*', the principal investigator spent prolonged time with potential participants before and after literacy classes, to establish a relationship of trust between the parties. She had several face-to-face discussions to explain the study's purpose, how it would be conducted, and answer any questions. As many people attending the *Association*'s activities have difficulty with written information, this strategy allowed potential participants to make an informed decision about their involvement. They were also all given an information letter about the study and the principal investigator's contact details, tailored for low literacy.

Data were collected using audio-recorded semi-structured interviews and a semi-structured, evolutive interview guide providing the flexibility to explore issues raised by participants. The interview guide was developed based on the Integrated Model of Health Literacy [3] and on the findings of previous qualitative studies on HL. Starting from their current health concerns or health problems, the participants' experiences of accessing, understanding, appraising, and applying health information were discussed. Health information needs, the difficulties or barriers experienced when dealing with health information, and strategies that were used to overcome these difficulties were also discussed. The interview guide is provided in Table 1. All interviews were conducted by the first author and principal investigator (CS) in a quiet, private room at the '*Reading and Writing Association*' or in the participants' home. Data collection stopped when data saturation was reached.

Before the start of each interview, participants self-administered a sociodemographic questionnaire including items on age, sex, marital status, nationality, educational attainment, occupation, income, social status, insurance status, and perceived health status. To describe the sample, each participant's HL level was also assessed using the validated French version of the 16-item, self-administered HLS-EU questionnaire [36], evaluating the experienced difficulties in accessing, understanding, appraising, and applying health information in the three domains of the health continuum by means of four-point Likert scales ranging from 'very easy' to 'very difficult'. Participants were assisted by the principal investigator if necessary. The scores per item were dichotomized by merging the 'very easy' and 'easy' scores into a score of 1, and the 'difficult' and 'very difficult' into a 0, and then summed. This produced a general HL score ranging between 0 and 16, with scores between 0 and 8 indicating inadequate HL, between 9 and 12, problematic HL, and between 13 and 16, sufficient HL [37]. Each participant was compensated with a CHF 15 (swiss francs) gift card for a local shop after completing the interview, to thank him/her for participating in the study.

The audio-recorded interviews were transcribed verbatim to ensure an accurate recording of participants' experiences. All the transcriptions were checked entirely for accuracy. Transcriptions were then entered into the data management software NVivo 11 (QSR International Pty Ltd, Doncaster, Victoria: 2016) for data storing, organizing, and coding. The data analysis process involved a hybrid method of inductive and deductive coding. Although the HLS-EU framework was first used to organize the data according to the four HL competence (Access, Understand, Appraise, Apply) [3] (deductive coding), the analysis and interpretation were conducted as inductively as possible and strongly connected to the data themselves (inductive

**Table 1. Interview guide.**

| Introductory question | |
|---|---|
| At the moment, what are your health concerns? | |
| **Main Framing Questions** | |
| What do you do when you have a health question and need health information? | |
| Framing questions by HL competences [3] | |
| ACCESS | What are your reasons for seeking health information? At what time, in what situation? |
| | What information sources do you use? |
| | How easy would you say it is to find/access/collect health information? |
| UNDERSTAND | Do you experience any difficulty in understanding heath information? In which context(s)? |
| | Would you say that health information is easy or difficult to understand? |
| | What do you do when you do not understand health information? |
| | What might help you understand better? |
| APPRAISE | To what extent do you trust the information found/received? For what reasons? What criteria are you using? |
| | How do you assess the credibility of health information? |
| | Do you experience any difficulty in appraising heath information? |
| | How do you deal with conflicting information? |
| APPLY | What do you do with the information you have found/received? How did you use it? |
| | To what extent does the health information you have found or received helps you to make health decisions? |
| | Would you say that health information is easy or difficult to apply/use? |
| | What might help you? |
| | When you make a healthcare decision, do you feel confident about that decision? |
| **Closing Question** | |
| Is there anything else you've thought of that you would like to mention? | |

coding). The thematic analysis was done according to the consecutive steps described by Clarke and Braun [31]: (1) data immersion: reading and re-reading the full transcripts to get an overview of the material to be analyzed and an overall understanding of the interviews; (2) generating initial codes: reading the transcripts in detail, line by line, to identify and extract the first relevant codes related to the research objectives. This was firstly done by hand, and then exported in the data management software; (3) searching for themes: grouping and organizing initial codes with common characteristics into thematic categories; (4) reviewing themes: refining the themes developed, examining their internal coherence, and generating a thematic map to explain the phenomena being studied; and (5) defining and naming themes: defining and describing the content of each theme in detail.

To address credibility [38], reflective journals, field notes and memos were continuously written down during the analysis, collecting impressions of each interview session, summarizing each interview and patterns appearing to emerge in the data collected. CS (a nurse researcher with expertise in the field of nursing science and public health) and AOB (a senior nurse researcher with expertise in the field of nursing and social sciences) worked together to analyze the data, and regular debriefing sessions were conducted to validate the themes, subthemes and categories identified as the process of analysis progressed. Any disagreements between the two researchers were resolved by discussion. To improve credibility of the analysis, the two researchers constantly reflected on the potential biases that they might carry due to their backgrounds, preconceptions, assumptions or previous beliefs. Authenticity of the results is supported by using the participants' own voices in the final report.

Ethical approval for this study was obtained from the Swiss Ethics Committees on Research involving Humans (Swissethics) (project-ID: 2017–00280). According to Swiss law, an ethical review was unnecessary for this qualitative research project not concerning human diseases or the structure and function of the human body, and collecting health-related personal data; this is explained in Articles 2 and 3 of the Federal Act on Research involving Human Beings [39]. The Ethics Committee did confirm, however, that the study was designed and planned following the general ethical principles applicable to any research involving individuals [40]. Before the study began, participants received all the necessary information, orally and in writing, and written informed consent was obtained from all participants.

This study is reported following the Consolidated criteria for reporting qualitative research (COREQ) checklist [41].

## Results

Twelve participants were recruited. Data were collected between April and June 2017, and interviews lasted from 30 to 150 minutes. Data saturation was reached after 10 participants. Two additional interviews were conducted to ensure data saturation. Participants' demographic and socioeconomic characteristics are shown in Table 2. Most participants were women, had completed compulsory school without continuing their education, were unemployed and receiving social integration benefits, and were covered by Switzerland's compulsory health insurance. Regarding their HL levels, four participants had inadequate HL, four had problematic HL, and four had sufficient HL.

The twelve interviews yielded rich data related to the strategies used by the participants when trying to access, understand, appraise, and apply health information across the three domains of the health continuum. Their description of how they proceeded when they needed health information, the barriers they encountered, and their strategies to overcome them, allowed to identify four main themes, and three main pathways regarding health information: (1) Financial insecurity triggers the need for health information; (2) Pathway 1: Physicians as ideal (but expensive) interlocutors; (3) Pathway 2: The internet as a suboptimal alternative; and (4) Pathway 3: Relatives as a default resource. In the next sections, the findings are presented in terms of these four themes and the subthemes that emerged from the interviews to emphasize their different facets (coding tree provided in Table 3). Our results show that the progression within the four HL skills is a complex process, with many steps backwards. Therefore, the three pathways are presented in a linear fashion for greater clarity, although they overlap. Additional participant quotations are also provided in S1 Table.

### Financial insecurity triggers the need for health information

**Recognizing the need for information to prevent unnecessary illness and save money.** Some participants indicated that they actively sought out information when confronted with a health promotion message that was out of step with their own lifestyle. The intention behind learning more about these messages was to adopt healthier, proactive behaviors and thereby improve their health. Other participants also mentioned the importance of learning about health to maintain or preserve their current health status. One reason was that they perceived health as something valuable or as a resource that needed to be preserved. Some participants who were facing great financial difficulties feared that becoming ill would result in unaffordable expenses (financial costs of treatment of the health problem coupled with a lack of income if unable to work). Such situations would not only affect themselves but also their dependents, such as children. Learning how to prevent illness by adopting healthier behaviors was seen as a means of not worsening an already precarious financial situation.

**Table 2. Demographic and socioeconomic characteristics of participants.**

| Participant | Gender | Nationality | Age | Education | Gross monthly income in CHF[a] | Employment status | Social Integration Income | Compulsory health insurance | Complementary health insurance | Self-perceived health status | HL score and HL level | Health topic addressed during the interview |
|---|---|---|---|---|---|---|---|---|---|---|---|---|
| 1 | M | CH | 44 | Vocational Education | 3000 | Unemployed | Yes | Yes | Yes | Good | 15 Sufficient | Small day-to-day health issues |
| 2 | W | CH | 50 | Compulsory School Completed | 2000 | Unemployed | Yes | Yes | No | Good | 13 Sufficient | Breast cancer |
| 3 | M | CH | 52 | Vocational Education | 1800 | Unemployed | Yes | Yes | No | Bad | 7 Inadequate | Chronic diseases—polyarthritis |
| 4 | W | NG | 44 | Compulsory School Completed | 3000 | Self-employed | No | Yes | Yes | Good | 11 Problematic | Chest pain |
| 5 | W | BR | 48 | Compulsory School Completed | 1580 | Part-time (40%) | No | Yes | No | Good | 11 Problematic | Small day-to-day health issues |
| 6 | W | CD | 48 | Compulsory School Not Completed | 1400 | Part-time (50%) | No | Yes | No | Bad | 7 Inadequate | Fatigue—muscle pain–treatment management |
| 7 | W | CH—BR | 45 | Compulsory School Completed | 3000 | Part-time (90%) | No | Yes | Yes | Very good | 15 Sufficient | Hypertension–prevention of complications |
| 8 | W | CH—CV | 57 | Compulsory School Not Completed | 3300 | Part-time (70%) | No | Yes | Yes | Good | 8 Inadequate | Urinary tract problems–treatment management |
| 9 | W | SO | 45 | Compulsory School Not Completed | 400 | Unemployed | Yes | Yes | No | Good | 6 Inadequate | Small day-to-day health issues |
| 10 | W | CH | 56 | Compulsory School Completed | 2000 | Unemployed | Yes | Yes | No | Good | 15 Sufficient | Menopause–adopting healthier behaviors |
| 11 | W | DO | 47 | Compulsory School Not Completed | 3000 | Unemployed | Yes | Yes | No | Bad | 12 Problematic | Adopting healthier behaviors–healthy eating |
| 12 | W | CH—IR | 60 | Vocational Education | 2000 | Unemployed | Yes | Yes | Yes | Good | 11 Problematic | Small day-to-day health issues |

BR = Brazil; CD = Democratic Republic of Congo; CH = Switzerland; CHF = Swiss francs; CV = Cap Vert; DO = the Dominican Republic; IR = Iran; M = man; NG = Nigeria; SO = Somalia; W = woman.

[a] At the time of data collection: 1 CHF = 0.87 € (Euro)/ 1 $ (Dollar)

**Table 3. Themes together with subthemes, HL competence and categories.**

| Themes | Subthemes | HL competence [3] | Categories |
|---|---|---|---|
| Financial insecurity triggers the need for health information | Recognizing the need for information to prevent unnecessary illness and save money | - | Being confronted with a health promotion message |
| | | | To preserve the current health status |
| | Finding the best way to treat the disease while maintaining financial stability | - | Having the financial means to cure the health problem |
| | | | Being responsible for one's own health |
| Pathway 1: Physicians as ideal (but expensive) interlocutors | Physicians as the main reference points for getting health information | ACCESS | The GP as the preferred source of health information |
| | | | The GP is the only identified source of health information |
| | Difficulty of obtaining health information due to limited access to a physician | ACCESS | The financial cost of a medical consultation cannot be afforded |
| | | | Long delays in getting an appointment |
| | | | Identifying alternative sources of information |
| | Insufficient information due to being disregarded or overly brief consultations | ACCESS | Not enough time during medical consultations |
| | | | Feeling disregarded |
| | | | Turning to other sources of health information |
| | Medical jargon makes information hard to understand | UNDERSTAND | Being confronted with unfamiliar medical jargon |
| | | | GPs' facilitative attitudes |
| | | | Asking for clarifications |
| | | | Not daring to ask more questions |
| | | | Health information recall difficulties |
| | | | Seeking additional information on the Internet |
| | Health information provided by GPs is not questioned | APPRAISE | Health information provided by physicians is trustworthy |
| | | | Conflicting information from different health professionals |
| | Striving to apply the medical treatment | APPLY | Too expensive medicines |
| | | | Discussing the treatment with the doctor |

(*Continued*)

**Table 3.** (Continued)

| Themes | Subthemes | HL competence [3] | Categories |
|---|---|---|---|
| Pathway 2: The internet as a suboptimal alternative | The internet as a tool to overcome barriers related to medical consultations | ACCESS | The internet as an alternative or compensatory strategy |
| | | | Internet's advantages |
| | | | Disincentives to using the internet |
| | The internet as a resource for day-to-day health issues | ACCESS | Seeking information for everyday health problems |
| | | | Seeking home remedies at no cost |
| | Hard-to-understand online information related to medical terminology and inadequately written information formats | UNDERSTAND | Difficult and unfamiliar words |
| | | | Difficulties in identifying key health messages |
| | | | Using a dictionary |
| | | | Turning to visual or auditory information |
| | | | Asking for help from family and friends |
| | Online health information is analyzed critically and viewed with suspicion | APPRAISE | Not all information is trustworthy on the internet |
| | | | Paying attention to the source of information |
| | | | Credibility assessment based on comments of other internet users |
| | | | Contradictory information |
| | | | Alarming information |
| | Online health information is selected on the basis of subjective and objective criteria | APPRAISE | Selecting understandable and brief information |
| | | | Relying on feelings and intuitions |
| | | | Selecting information based on beliefs and values |
| | | | Information selection based on financial and material means |
| | Using simple, easy-to-implement online health information | APPLY | Implementation of small tricks at no cost |
| | | | Lack of financial means |
| | | | Importance of motivation to adopt new health behaviors |
| | | | Evaluating the effectiveness of home remedies |
| | Spreading effective, small everyday health suggestions and tricks to family and friends | APPLY | Sharing experiences with others |
| | Using online health information to discuss and negotiate with the doctor | APPLY | Negotiating therapeutic alternatives with the doctor |
| | | | Discussing online health information with the doctor |
| | | | Feeling better prepared for care |

(*Continued*)

**Table 3.** (Continued)

| Themes | Subthemes | HL competence [3] | Categories |
|---|---|---|---|
| Pathway 3: Relatives as a default resource | Trusted relatives are asked for health information | ACCESS | Asking health advice to trusted family members or friends |
| | | | Social network members familiar with the health topic |
| | | | Individuals with expertise in the field of health |
| | | | Confidentiality |
| | Trusted relatives have no answer | ACCESS | Relatives have the same difficulties with health information |
| | Relaying effective advice from relatives | APPLY | Becoming health information relayers |

*"If you get sick, you have to go to hospital, but it's expensive. Sometimes you have to pay a certain amount of money before they start the treatment. And then there's my whole family, counting on me. What am I going to do if I get sick? So, I take it easy. So, I said, 'Let's not be ignorant. Let's make an effort not to get too sick [. . .], avoid unnecessary illness. [. . .] Because I'm not rich: I'm always paying out. So, if I have my health, everything is fine."*

*(Participant 4)*

**Finding the best way to treat the disease while maintaining financial stability.** Health information needs also emerged when people were faced with an illness they did not know and about which many questions arose. Participants mentioned the importance of getting answers that would help them understand what their problem was and that would remove uncertainty. Actively seeking health information was motivated by issues such as having the financial means to cure the health problem or wanting to avoid potential complications. Participants sought information so that they could take action and be responsible for their own health, make health decisions and achieve well-being.

*"At first, honestly, I felt powerless. [. . .] You get angry because you don't understand why; you don't know what's happening to you. It would be good for doctors to realize that we are sick, that we are not just numbers and a wallet! So, now [. . .] I have to know how I can treat my disease!"*

*(Participant 3)*

## Pathway 1: Physicians as ideal (but expensive) interlocutors

**Physicians as the main reference points for getting health information.** All the participants identified their physician, usually their general practitioner (GP), as their preferred source of health information. GPs were identified as the most competent professional to ask about health issues, a reference figure, and as 'the one who knows'.

*"But the doctor is always my first help. That's where I start."*

*(Participant 9)*

For some participants, their physician was their only identified source of health information, and some said they would not know where else to turn to.

*"I don't know anything about asthma. And that annoys me! And I don't know where to go to get any information about it [besides the doctor] . . ."*

*(Participant 4)*

**Difficulty of obtaining health information due to limited access to a physician.** Although all the participants primarily cited their physicians as their main health information source, the financial cost of a medical consultation was, for some, a real barrier to using this source.

*"And sometimes I really don't feel like going to the doctor all the time either. That's because when the bill comes, we don't exactly throw a party at our place! We have to pay afterwards, that's what worries me."*

*(Participant 8)*

Faced with a precarious financial situation, some participants expressed they only go to the doctor if the health problem is deemed to be serious or too disabling.

*"When I had this mobbing problem, I didn't think about how much it would cost [to go to the doctor]. When you get to a point like that in relation to your own health, the barriers come down."*

*(Participant 1)*

Participants also stated the importance of obtaining quick answers to their health questions. However, some participants identified the very long delays in getting an appointment or the long gaps between medical consultations as barriers to getting information from their doctor.

*"The problem is that he [the doctor] doesn't give you an appointment within the week. He gives you one in two weeks or three weeks. . . But you need help now, so you find it at home [using the internet]."*

*(Participant 9)*

All these situations made participants feel quite helpless, leading them to identify other potential health information sources, such as the internet, or friends and family. Used as a compensatory strategy, consulting these alternative sources of information enabled participants to obtain health information without increasing their financial difficulties.

*"Nowadays, if something happens to you [a health problem], you'll call your friends and all that. That costs nothing. You don't have to pay for that. . ."*

*(Participant 9)*

*"I found the intervals between treatments long—there was lots of waiting. [. . .] With this disease, you're always. . . at the slightest problem, you ask yourself questions. [. . .] So, it's true that I used to go on the internet a lot."*

*(Participant 2)*

Some participants also mentioned turning to their pharmacist, for multiple reasons: pharmacists are recognized health professionals with a university degree; they are immediately accessible, with no need for an appointment; they can be consulted free of charge; and they are geographically accessible, as there is always a pharmacy near one's home.

> *"Sometimes, it's true; you don't have to go to the doctor. You can go to the pharmacy, and they'll explain it to you. It can still help sometimes. Yes, because you can't go to the doctor all the time when you have a question."*
>
> *(Participant 12)*

**Insufficient information due to being disregarded or overly brief consultations.**
Among the participants who had access to their physician, some reported that they did not always get the information that they wanted when consulting their doctor. Many remarked that they did not receive enough information or explanations about their health issues. They perceived the information as too superficial and felt that they were often unable to ask all the questions they wanted to ask. This could be for different reasons. Participants were aware that physicians see many patients and that the time available for asking questions and getting detailed information is limited or insufficient. Participants also thought that doctors sometimes fail to listen to them, that they are disregarded, and act as if they are not open to discussion and communication. In some cases, physicians voluntarily transmitted only superficial information to avoid worrying their patients unnecessarily.

> *"Well, I missed out on it a little bit, yes, because I had so many questions to ask him [the doctor]. [. . .] I don't have all the answers to all the questions I've got. [. . .] They [physicians] don't have the time anymore, they've got to be productive; and then it's financial, it's only financial."*
>
> *(Participant 3)*

To overcome these obstacles, participants turned to other sources (usually the internet) to compensate for the lack of information received from their doctor.

> *"It annoys me that the family doctor says, 'Aha! You have a blood pressure problem'. [. . .] And I've never had much explanation about it. He's never explained to me where it comes from. And that really bothers me, [. . .] that's why I tried to do it the other way [via the internet]."*
>
> *(Participant 7)*

**Medical jargon makes information hard to understand.**   Participants mentioned difficulties in understanding the information they received during medical consultations. The main obstacle declared was being confronted with unfamiliar medical jargon, which led to misinterpretations of messages. Faced with difficulties in comprehension, participants pointed out how they felt lost during interactions with their doctor and consequently ended up with even more questions.

> *"But just like with the doctor, there are many languages, many parameters, . . . things that I don't understand at all."*
>
> *(Participant 8)*

When facing barriers to understanding information from doctors, participants could adopt various strategies to overcome them. Some asked for clarification or for misunderstood information to be rephrased.

However, others admitted to feeling embarrassed to express their misunderstanding. Some feared that they were taking up too much of their physician's time, or perceived their doctor as an authority figure in terms of health knowledge Not daring to ask more questions was also linked to feelings of shame related to a lower educational level. These participants felt 'incompetent' in front of their doctor and reported difficulties in clearly expressing their concerns and finding the appropriate words to express that they did not understand.

*"I don't really understand the doctor. And sometimes I feel embarrassed [to tell him]. [. . .] I never try to ask more questions. . . I didn't know. I feel embarrassed. . . I don't have the knowledge! I don't know. . . And I can't find the words to explain it."*

*(Participant 8)*

Moreover, the use of complicated words or overly 'medical' jargon prevented some participants from being able to recall the information they were given.

*"I don't feel at ease at the doctor's. . . [. . .] Because I've got to memorize all this, and I just can't get it into my head. So, it's twice as complicated because there are things I don't understand, and because there are things I have more trouble remembering."*

*(Participant 8)*

Taking notes was one strategy suggested for remembering the doctors' explanations, as was searching for additional information on the internet.

*"I don't have a good memory. [. . .] At first, I didn't really know what kind of cancer it was, so. . . BRC 2 RC. . . what's that? Wait, I'm writing this down. [. . .] It's true that afterwards I went on the internet and I typed in the name of my cancer. So that's how I knew why I was given [this medical treatment]."*

*(Participant 2)*

If problems of incomprehension persisted after a medical consultation, many participants turned to other sources of information, primarily the internet, to seek additional information and to try to better understand the points raised in their discussion with the physician.

**Health information provided by GPs is not questioned.** In general, health information provided by physicians was considered trustworthy. Doctors are recognized as health experts and a trusting relationship had often been established for years.

*"I trust [the doctor] because it's his field, he knows what he's saying, he's not going to bullshit me."*

*(Participant 4)*

However, difficulties assessing the credibility of health information arose when participants were asked to consult multiple physicians or other health professionals. Several explained that they had met with conflicting information from the different professionals they had consulted. Some participants found this very problematic, creating confusion and doubts about the

information received and the behaviors to adopt. Faced with this, participants prioritized information given by their consulting physician (mainly their GP) because of the more well-established, trusting relationship.

> *"[Conflicting information from doctors] makes me ask questions... [...] I'll always choose my general practitioner, the family doctor, that's it. I've been with her for more than 14 years. I trust her most because she knows everything, she has my entire record."*

> *(Participant 8)*

**Striving to apply the medical treatment.**   As the main reason for consulting a doctor was to solve a health problem, applying the health information that was provided during the consultation was essentially limited to correctly taking the treatment that was prescribed.

Some participants expressed difficulties in correctly taking the treatment prescribed by the doctor due to the high cost of the medicines. As a result, they explained trying to get their medicines abroad because they are less expensive.

> *"There are different medications I need to take. I went to buy them in France because they are much cheaper! [...] So yeah, I don't agree with this system. It's so expensive to be treated!"*

> *(Participant 10)*

Some participants mentioned critically evaluating the effectiveness of their treatment; if it was ineffective, it was discussed and negotiated with the doctor to find an alternative. However, some participants encountered unresponsive physicians. In that case, they turned to the internet for other means and solutions for self-treating their health problem.

> *"I turn to the doctor; I look at what he prescribes me. If the medication works, that's it. If it doesn't work, that's when I look on the Internet!"*

> *(Participant 9)*

### Pathway 2: The internet as a suboptimal alternative

**The internet as a tool to overcome barriers related to medical consultations.**   Participants consistently and widely used the internet, either as an alternative or compensatory strategy when they were unable to consult their doctor (due to financial and time constraints), or as a complementary strategy to medical consultations when those did not provide sufficient answers to their questions. The internet's main advantage is that it is a 'free' source of information requiring no additional financial expenses, immediately and permanently accessible, as opposed to the waiting times for a doctor's appointment which can be very long.

> *"So, on the internet, I don't have to pay, and in one way, it's faster. Yes, money's a problem. It's always about money... It's cheaper too."*

> *(Participant 9)*

Despite its accessibility, some participants mentioned that internet is not really free: financial costs (an internet subscription and the cost of a computer) are barriers to using the internet as a source of health information.

*"No, on the internet? I can't because I don't have internet on my phone. I don't have it because it costs too much. And I'm trying to keep costs down a bit."*

*(Participant 4)*

Some participants expressed difficulties using computer tools correctly, explaining that their use required skills that needed to be learned.

Several participants also mentioned that actively searching for health information on the internet is a process requiring a significant investment of time, which is a resource that is not always available in a precarious socioeconomic situation or under difficult living conditions (e.g., working two jobs to survive, seeking employment, exhaustion).

*"I never go on the internet for health reasons. I'd need time for that."*

*(Participant 5)*

**The internet as a resource for day-to-day health issues.** Most participants used the internet to solve small, everyday health problems, seeking tips, home remedies, or traditional remedies. The reasons for seeking this type of information were mainly financial. With limited financial resources, they sought solutions that are easy to apply or implement with everyday products (found at home) and that help avoid having to spend money for treatment.

*"It's true that the internet is a great place to form your opinion. Just to find tips and tricks to take care of yourself. [. . .] I try to see if I can find a solution on my own."*

*(Participant 10)*

**Hard-to-understand online information related to medical terminology and inadequately written information formats.** Participants identified several factors that created barriers to their understanding of online health information. Many reported difficulties in making sense of the health information they found when viewing webpages of written information, particularly regarding the terminology used and the many new and unfamiliar words that hindered their overall understanding of the text.

*"And even if you speak French well, the explanations use words that you don't hear very often, and that's what blocks me."*

*(Participant 9)*

Participants also reported that lengthy texts made it difficult to identify the key health messages. Many said they preferred short, clearly written information with the essential information immediately identifiable.

*"Sometimes there are long explanations, and then sometimes there are [only] two sentences, three sentences, and you understand better."*

*(Participant 9)*

These two barriers led some participants to stop using the internet to actively search for health information. The difficulties in understanding written information were sometimes so great that they no longer wanted to use the internet to find health information.

*"But I must admit that the way I do things on the internet, if I don't understand it, if I get fed up, I stop."*

*(Participant 10)*

However, some participants developed strategies to overcome these difficulties. Barriers related to medical jargon, or complicated words were overcome by using a dictionary.

*"It could be I find words whose meaning I don't know. So, I'll note it down and look in the dictionary because I've a dictionary at home that helps me."*

*(Participant 11)*

As extensive written information often provided problems, many participants preferred illustrated information or more visual and auditory information, such as YouTube videos.

*"Well, going on Google, there are a lot of videos. So I watched those videos a lot. When it says there are six pages to read... Then, I'll switch to a video... For me, the tablet, Google, it was great. That's where I understand... because there are lots of drawings, lots of things, videos, reports from other doctors on YouTube..."*

*(Participant 2)*

Another strategy for overcoming barriers was using trusted people in one's close social network to ask the meaning of words and to increase their understanding of the information they had found. Several participants in this study were recruited through an adult literacy center, and some sought help from their teacher when they found information on the internet difficult to understand.

**Online health information is analyzed critically and viewed with suspicion.** All the participants expressed doubts about the reliability of the online health information they found. They were aware that not all information is trustworthy and, therefore, adopted a cautious attitude, although they expressed their difficulties to distinguish 'real' information from 'fake'.

*"On the internet, the problem's also that you can't be 100% certain of the information."*

*(Participant 1)*

Faced with this difficulty, some participants reported feeling lost and confused. When in doubt about the quality of health information, some consulted several different web pages on the same health topic and triangulated their information to confirm or refute it.

*"And then if there are things that I trust, I'll take some keywords and look further, to confirm or reject what was written."*

*(Participant 10)*

To assess the quality of online health information, a few participants reported paying attention to the source and used several criteria to attest to the credibility of the information, such

as whether the website was reputable, professional, of a recognized health institution (such as a hospital), the author's notoriety or professional status (e.g., being a physician), or the popularity of the website accessed.

> *"Because I only see that doctor on YouTube. He's very well known, he's done lots of books, lots of publications. He gives courses for doctors, and he gives courses at the University. He has done a lot of training. He works a lot with television stations in Brazil. He does loads of things, so. . . Very famous!"*
>
> *(Participant 7)*

Some participants also tried to determine the reliability of their information by looking at the opinions, comments, and criticisms of other internet users. For YouTube videos especially, the number of views was also an element contributing to the credibility assessment.

> *"And I also look at the rankings of the videos [the number of views]. How many people have watched. I go on that a little bit, too. And that's why I trust it a little bit. . ."*
>
> *(Participant 7)*

With regard to the content of the online information, two main difficulties were mentioned. Participants explained that they frequently found contradictory information on different websites or information described in different words, which led to confusion and feelings of being lost.

> *"That's why I don't trust it, and I don't go on the internet, because sometimes you see a piece of information here, and then you see the same information using other words. Sometimes it confuses your mind. . . It confuses my mind a lot. . ."*
>
> *(Participant 5)*

Participants also explained that an internet search for a trivial symptom or health concern could lead to finding alarming information, causing feelings of worry, scare, or anxiety about the severity of their health problem.

> *"I went on the internet about a stomach nodule. Again, it's cancer, it's immediately cancer, it's immediately unbelievable garbage. I said, 'No, stop!' I stopped searching."*
>
> *(Participant 3)*

Faced with these problems, different attitudes were adopted. Some participants chose to ignore internet pages with conflicting information, others stopped searching for health information on the internet, and some would never use the internet as a source of health information again. Others continued to use the internet but turned to other sources such as trusted individuals with experience on the topic to clarify the information they found online and help them select reliable information.

> *"And then when there are articles that say the opposite, [. . .] I read them too. It's a bit like the elections: you don't know what to think (. . .) Afterwards, I'll go and get information from professionals whom I know, or people who know a little bit about the subject."*
>
> *(Participant 10)*

**Online health information is selected on the basis of subjective and objective criteria.**
Given the difficulties to assess the credibility and reliability of online health information,
participants described several criteria they used to select reliable information. Some
explained that health information had to be presented clearly, understandably, and
briefly.

*"If it grabs my attention, then it's because it's concise, it's clear. . ."*

*(Participant 10)*

Health information was selected when it was easy to remember, with some participants
mentioning that it had to "look serious", i.e., contain scientific vocabulary, which may seem
inconsistent with the difficulties associated with understanding medical jargon.

*"To trust it, . . . well, it's about how the article's written. There have to be some slightly scientific words; it has to look a bit serious."*

*(Participant 10)*

Other subjective mechanisms for selecting health information used by many participants
was to rely on their feelings or intuitions about the content.

*"How do I select information? Well. . . I always trust my intuition. Because I find that my intuition doesn't let me down, it's great."*

*(Participant 3)*

Information also tended to be more easily selected if it was consistent with participants'
beliefs, convictions, preferences, and values.

*"I select the information according to my convictions too, because for me, anything chemical is not great, it's really not good."*

*(Participant 10)*

However, an important selection criterion was having the financial and material resources
to follow health recommendations. Many participants explained that the health recommendations they selected were those that did not involve significant financial costs (e.g., using a
home remedy made from common household products) and which they thought would be
easy to implement.

*"So, when a list comes up [when I search on the internet], I look for what I can find in my house, that I don't need to go and buy. I really avoid buying stuff."*

*(Participant 12)*

**Using simple, easy-to-implement online health information.** After having sought and
select health information on the internet, participants applied those messages that did not
require significant amounts of time and money. The information that was mostly implemented consisted essentially of one-off things or small, everyday health suggestions and tricks
that they could apply on their own.

*"If it's easy to do, then I'll do it if they're small things. Things I can do by myself, alone; things I can do right away, if it's not complicated."*

*(Participant 12)*

Information on healthy lifestyles, particularly involving prevention and health promotion requiring longer-term behavioral changes or financial means, was more difficult to implement and sustain.

*"I wanted to change the way I eat, to be healthier. . . But before, there were two of us at home, we were paying both for the groceries. And now I'm all alone, I can't afford it by myself. So I can't do it anymore. . . [eating healthy]."*

*(Participant 4)*

Some participants mentioned the importance of motivation and the willingness to adopt new, more positive health behaviors over time, explaining that motivation can be negatively affected by a socioeconomically disadvantaged situation because of the anxiety and uncertainty this creates. Some participants mentioned the importance of having the self-confidence to be able to correctly apply online health information.

*"It's hard to implement. . . because [. . .] I don't have much willpower. But I kept my energy levels up, and I lost 8 kilos. But I've started to drop off again. . . [laughs]. But when you're not in your right mind, when you don't feel stable and you want to find a job, you're not right. . ."*

*(Participant 11)*

Participants explained that they evaluated the effects on their health of implementing home remedies. Because the implementation of such remedies was conditioned by the availability of financial resources, the expected results were not always achieved. If they proved ineffective, and participants were thus unable to manage their health problem alone, some resigned themselves to consulting a healthcare professional despite their financial difficulties. One participant, however, had developed a specific strategy to avoid unnecessary expenses by turning to the pharmacist for advice on whether to consult her doctor to solve her health problem.

*"So if the home remedy doesn't work, then I go to the pharmacy first. [. . .] I ask for my pharmacist's opinion."*

*(Participant 12)*

**Spreading effective, small everyday health suggestions and tricks to family and friends.** If participants deemed the solution implemented effective, they shared their experience with family and others by becoming 'messengers' or vectors of health information themselves.

*"You try it; when you see that it works, then you go and give advice to others. But I don't give advice if I haven't tried it first. [. . .] You should never give advice if you haven't tried it. Maybe the person will get sick, and then it's your fault."*

*(Participant 9)*

**Using online health information to discuss and negotiate with the doctor.** Participants who faced a major health problem often used the health information they found online to complement their medical consultation, as something to be discussed during the next meeting with their doctor. Therapeutic alternatives identified on the internet could then be negotiated.

*"So they were talking about polyarthritis on the internet. [. . .] You can have an injection once a year, and then you're okay all the time. Why wasn't I offered these treatments? Why, well. . . I'm going to ask my doctor again. . ."*

*(Participant 3)*

Discussing online health information with the doctor was also a means of validating its quality and thereby enabled participants to increase their confidence in their own ability to act.

*"For health. . .well, you can use the internet now. But you can't really trust it 100%. You should still ask your family doctor questions. You can't do anything risky. . ."*

*(Participant 7)*

Online health information was considered a means to play a more active part in health decision-making as a patient, of being well informed, and of being able to truly interact with health professionals. It also helped participants to feel more comfortable, reassured, and calm about committing to a care pathway and to deconstruct some of the erroneous beliefs related to their illness.

*"So, when I went on the internet to look up chemotherapy, [. . .] I knew more or less what to expect. . . it prepared me to be less afraid. . . because I didn't really know where I was going with these chemotherapies."*

*(Participant 2)*

## Pathway 3: Relatives as a default resource

Like the internet, using one's social contacts was seen as a compensatory or alternative strategy. When financial barriers made it impossible to consult a doctor and participants had significant difficulties using the internet, turning to friends and family was a way to obtain health information and advice for free.

**Trusted relatives are asked for health information.** The members of participant's social networks involved in HL processes were most often nuclear or close family members and friends. Work colleagues or clients were also identified as health information sources, with health advice and information sought mainly in trusted relationships, from individuals with expertise in the field of health or familiar with the health topic in question, or from whose who had undergone the same experience.

*"I'd call my father, in Canada; his wife is a nurse. And then, as they're quite. . . I'd say. . . in that field, it's true that I asked my step-mother a lot of questions."*

*(Participant 2)*

Confidentiality was seen as an important factor, with some participants rejecting the idea of discussing health issues or concerns with people around them whom they did not fully trust, for fear that their health concerns would leak.

*"I don't ask my friends for health advice! I don't trust that; I don't like it. If I tell you something, it can't go any further. I don't like something private going any further. If we talk, it has to stay between us! I don't like that. I don't want that."*

*(Participant 8)*

When participants did get information and advice from family and friends, its quality was not questioned and the advice given was implemented almost immediately.

*"Now, when I have a health problem, my girlfriends are there. So, I called a friend; I said 'I've got this pain. What's happening to me? I'd like to know.' Then she said, 'Take this.' I said yes."*

*(Participant 9)*

**Trusted relatives have no answer.** Although family and friends were often sources of support and advice, some participants found it difficult to obtain health information from those around them. Several participants noted that members of their network often faced the same difficulties with health information as they did, did not have the answers to their questions, and advised them to see a doctor to get answers.

*"My family is my brother, and the families I've started. So, they say, 'Go see the doctor,' and that's it. They're no more experienced than I am. It's hard, because even they don't know."*

*(Participant 9)*

**Relaying effective advice from relatives.** Participants stated that they evaluate the effectiveness of the advice that they received from others in the same way they did with online health information. When found to be effective and relevant, they became health information relayers themselves.

*"So, I was able to pass the information on to others who'd gone through it, because there were some who didn't know how."*

*(Participant 2)*

## Discussion

The present study aimed to explore the experiences of socioeconomically disadvantaged people in accessing, understanding, appraising, and applying health information. The qualitative analysis confirmed that HL is a process which socioeconomically disadvantaged people perceive to be an 'obstacle course', because they encounter many difficulties.

Our results revealed three main barriers regarding health information: financial deprivation, barriers to obtaining and understanding health information during the medical consultation, and the lack of digital skills when using the internet for health information.

Financial deprivation, a recurring aspect in the interviews, seems to be the most important barrier to HL in Switzerland. Several studies have shown that financial deprivation is one of the major factors leading to low HL levels [7, 8, 42]. In our study, this link between financial deprivation and low HL could be further specified by showing that a lack of financial resources limits the access to the healthcare system (the high financial cost of medical consultations) [43] and, to a lesser extent, to online health information. The way Switzerland's healthcare system is organized and financed represents a barrier to the use of its resources and to the access to healthcare professionals as sources of health information, and therefore contributes to limiting the development of HL skills among the socioeconomically disadvantaged, who need it most.

Our findings showed that socioeconomically disadvantaged people turn to the internet as a compensatory strategy, mainly to find self-treatment solutions or inexpensive home remedies. As such, it reflects the poor access to healthcare of people with a precarious or disadvantaged socioeconomic status in the Swiss context, due to the financial difficulties and a poorly developed primary healthcare system in Switzerland [44, 45].

During their medical consultations, many participants in our study mentioned various barriers to obtaining health information (e.g. having insufficient time to ask questions) and understanding health information (e.g. being unfamiliar with medical jargon or difficulties in remembering their doctor's instructions). Studies among disadvantaged populations have shown that only half of the information conveyed during a medical consultation is understood and recalled [46, 47], and that 40% to 80% of the information is forgotten immediately [48]. As such, physicians' attitudes and skills seem to be a key to helping socioeconomically disadvantaged people overcome the challenges of low HL during medical consultations [49]. Good communication skills of physicians are associated with better health outcomes among patients [50]. Conversely, poor patient–physician communication contributes to poor health outcomes [50] and is considered a factor that contributes to health disparities [51]. The quality of doctor–patient communication follows a social gradient, in the way that patients from lower socioeconomic backgrounds receive less health information, fewer explanations, less listening time, less advice, and shorter consultations than patients from higher social classes [49]. Several tools or techniques, known as universal precautions, can be used to ensure patient comprehension, encourage them to ask questions, and improve health-information recall [52]. These include for example avoiding medical jargon, repeating and summarizing instructions, giving a limited number of instructions at a time, using pictograms or illustrations to complement oral information, and using the teach-back approach [50, 52, 53]. The latter is a known and promising technique for promoting understanding and recall [50]. However, only 39.5% of healthcare professionals used this approach regularly [54]. To enhance the use of these strategies, interventions are necessary that aim to improve health professionals' communication skills, encourage a more patient-oriented approach [51], and create a more supportive organizational environment. Curricula for the training of healthcare professionals should incorporate these communication techniques as a universal means to optimize patients' understanding of the health information they receive [53]. According to the universalist approach, all patients will benefit from this type of strategy, regardless of whether they have high or low HL and a high or low socioeconomic status [51].

Using the internet was the most common compensatory strategy mentioned by the participants in our study to overcome barriers related to medical consultations. Our participants described various difficulties in understanding online health information, such as medical jargon, key messages that are not immediately identifiable, and overly long texts. The complexity of online information is well documented: whereas the US Department of Health and Human Services suggests that health information should be written at or below the level for 11- to 12-year-old children (after six years of education) [55], numerous studies have shown that

online health information is written at a level two to six years of schooling above this recommendation [56–60]. So, although the internet provides access to vast amounts of health information, it is not always helpful to the socioeconomically disadvantaged. In fact, the complexity of online health information may even contribute to health inequalities [58]. To be a useful health information source for the socioeconomically disadvantaged, online health information should be presented at an appropriate reading level [58]. This would require, first of all, the use of plain language, i.e., a communication style which '*use language, structure, and design so clearly and effectively that the audience has the best possible chance of readily finding what they need, understanding it, and using it*' [61]. Secondly, combining plain language with illustrations can be a strategy to improve the comprehension of online health information. Developing easy-to-read, easy-to-understand, and easy-to-navigate online health information should thus be a priority [62, 63]. This will not only benefit people with low HL levels but also those with adequate HL skills.

Appraising online health information appeared to be the most problematic HL skill for our participants. Many of our participants discovered conflicting or alarming online health information, that, as pointed out by Synnot et al. [64], can create skepticism regarding the information that is found, and an inability to determine whether it is reliable. Conflicting information creates confusion, uncertainty, fear, and anxiety [65], decreases self-efficacy [66], and negatively impacts health decisions, health behaviors, and health-related outcomes [67–69]. Consequently, participants who experience difficulties in sorting and selecting online health information resort to the source and information which they feel matches their values and preconceptions best [65, 70]. All our participants were aware of the risk of online health information and did not trust it, yet some chose to use online information despite this lack of trust. This is troubling because selecting 'bad' health information, particularly if coupled with problems of comprehension, can lead to misinterpretation, negatively affect decision-making processes, and result in negative health consequences. So, because accessing health information is easy, information technology has the ability to empower healthcare consumers and to make people better informed about health topics, enabling them to make better health decisions, participate more actively, and achieve better health outcomes [71]. However, at the same time, our findings indicate that the internet may also deepen existing health disparities or create new ones [56, 72, 73]. This is because socioeconomically disadvantaged people, who are more likely to have poorer HL, may experience disproportionately greater difficulties in understanding and appraising online health information than more advantaged individuals [71, 74, 75]. It would therefore be important to understand the online appraisal skills of socioeconomically disadvantaged people in more detail, as a basis for developing new skills-based interventions that enable them to effectively use the internet and critically evaluate online health information [76]. However, there are not many well-documented evidence-based strategies to help people appraise online health information [65]. Although there is an abundance of institutional websites directing healthcare consumers to guidelines or tools for critically evaluating the quality of written health information, and effect studies have shown that some of these tools could improve participants' critical skills regarding health information [77, 78], these studies were conducted in a predominantly well-educated population. Little is known about their effectiveness in improving the critical skills of socioeconomically disadvantaged people. Although health professionals could refer their patients to such tools [79] or recommend accurate websites [80], this strategy is constrained by the need for contact with a healthcare professional, while disadvantaged people are often face financial barriers to consult them. Furthermore, this strategy is not useful for people who consult the internet to self-diagnose and treat health problems alone, without consulting a healthcare professional [67]. Other possibilities would be for community-based organizations to transmit these tools to their members [73, 81], or to teach

critical HL, as well as digital skills at school [82, 83]. Such interventions would not only benefit the socioeconomically disadvantaged but also all future generations.

Overall, our participants applied very little of the online health information they accessed, probably because they did not have sufficient prerequisites (understanding and appraisal skills) to apply that information adequately. However, the health information they had accessed online did help persons with chronic conditions who had regular contacts with their physician to better discuss their disease and negotiate treatment aspects during their medical consultations. This concurs with findings from other studies [67, 84–86] asserting that online health information empowers people '*to do something rather than just being told what to do*' [85]. It allows them to take up a more active role in the decisions concerning their health. Curiously, one thing participants did with online health information was share it within one's social networks. We hypothesize that the dissemination of health information is motivated by a desire to assist others experiencing the same financial barriers to accessing healthcare or information, as well as the same difficulties in understanding, and appraising health information.

Our results also suggested that the members of social networks can be important HL resources. According to the notion of 'distributed HL', social network members act as HL mediators, providing support regarding health information, and compensating for '*personal deficit in HL skills*' [87]. However, seeking advice or support from fellow social network members also revealed certain barriers in our study. Indeed, family members or friends had difficulties providing the information our participants wanted. As shown in a previous study [88], we hypothesize that people from the same social environment share the same socioeconomic characteristics, the same HL levels, and consequently, the same difficulties regarding health information.

### Limitations and strengths

This study is not without limitations. Firstly, the sample size is relatively small. This can be explained by the fact that, as shown by some previous research, it is difficult to access and engage socioeconomically disadvantaged groups in research projects. Socioeconomically disadvantaged people are therefore considered a 'hard-to-reach' population [89, 90]. Consequently, the results could not be contrasted according to the participants' level of HL. Other experiences or additional difficulties with health information could have been highlighted if more participants with a problematic or insufficient HL level had been included. Secondly, most of our participants were of immigrant origin and not very fluent in French (the language in which the interviews took place). This language barrier may have exacerbated the difficulties they encountered with health information and limited the richness of the data collected. For this reason, our sample may not be fully representative of Switzerland's socioeconomically disadvantaged population. Thirdly, most of our participants were recruited from adult education programs. It is possible that they had already received some training on how to overcome some of the barriers relating to information in general and that some of the compensatory strategies we have mentioned were taught to them during literacy classes. Fourthly, most of our participants were women. Several studies have shown that women have slightly higher HL levels than men [91, 92]. This may be because women report health issues more frequently than men [91]. Women are also the most involved in the health of their families, especially their children. As a result, women use the healthcare system more frequently and are thus more familiar in navigating the healthcare environment [91]. In addition, women are more likely to use the internet for searching health information than men [93], and have thus a better knowledge of online health resources. Fifthly, the age range of the sample is rather narrow (age range 44–60). This can be explained by the recruitment sites chosen, which are specifically

aimed at the working population. This may have limited the richness of the data regarding digital HL skills and the use of information technologies (e.g., the internet). Indeed, the difficulties and barriers related to digital skills might have been greater if more older people had been included in the sample. And finally, the data was collected in 2017. Since then, the COVID-19 pandemic has significantly changed the way people access health information. People with a low level of education, low income or belonging to an ethnic minority were disproportionately affected by COVID-19 [94, 95], probably because the COVID-19 pandemic was itself accompanied by a phenomenon called 'infodemic', defined as an overabundance of health information of varying quality, including the dissemination of fake, inaccurate, incomplete, unverified, contradictory, or misleading information [95, 96]. This has led to confusion and exacerbated difficulties in understanding, appraising, and applying health information, especially among people with low levels of HL [96]. Consequently, the barriers and strategies described by our participants may only reflect some of the real difficulties encountered by socioeconomically disadvantaged people with health information.

Despite these limitations, we believe that this study has a number of important strengths. One of its main strengths is its qualitative approach, which enabled an in-depth exploration of HL skills from the perspective of socioeconomically disadvantaged people themselves. The participants in this study had different HL levels, cultural backgrounds, health concerns, and health information needs, which allowed to highlight the range of difficulties and barriers that socioeconomically disadvantaged people encounter with respect to health information, across the entire health continuum (healthcare, disease prevention, and health promotion). Another strength is that we explored the concept of HL as an asset, rather than as a deficit [4, 12]. Finally, the data collection and analysis were guided by the Integrated Health Literacy framework [3], which has been little used in qualitative research to date. Using this framework helped us contribute new perspectives to the fields of HL and health disparities.

## Conclusion

Socioeconomically disadvantaged people have a strong desire to manage their own health and engage in behaviors that will satisfy their health information needs. Their progression in the HL process is like an 'obstacle course', with numerous difficulties that must be overcome to access, understand, appraise and apply health information. To that effect, these people often resort to compensatory strategies. If these strategies are ineffective, the barriers leave individuals with unanswered health questions. Financial deprivation is the most important barrier to HL in Switzerland, preventing people from accessing both healthcare and online health information, and influencing their selection and application of health information, recommendations, and guidelines. Appraising health information is the HL skill with which socioeconomically disadvantaged people struggle the most. A range of physician-based, individual skills-based, organizational, and policy-based interventions are needed to help socioeconomically disadvantaged people overcome their HL challenges.

## Supporting information

**S1 Table. Additional participant quotations.**
(DOCX)

## Acknowledgments

The authors would like to thank all the participants who participated in the interviews, as well as our community partners (Association « Lire et Ecrire », section Lausanne et Région,

Lausanne, Switzerland; and Office Régional de Placement, Châtel-St-Denis, Switzerland). The researchers would like to acknowledge Darren Hart for the English translation.

## Author Contributions

**Conceptualization:** Coraline Stormacq, Stephan Van den Broucke.

**Data curation:** Coraline Stormacq.

**Formal analysis:** Coraline Stormacq, Annie Oulevey Bachmann.

**Investigation:** Coraline Stormacq.

**Methodology:** Coraline Stormacq, Stephan Van den Broucke.

**Project administration:** Coraline Stormacq.

**Resources:** Coraline Stormacq.

**Supervision:** Stephan Van den Broucke, Patrick Bodenmann.

**Validation:** Annie Oulevey Bachmann, Stephan Van den Broucke, Patrick Bodenmann.

**Visualization:** Coraline Stormacq, Annie Oulevey Bachmann.

**Writing – original draft:** Coraline Stormacq, Annie Oulevey Bachmann.

**Writing – review & editing:** Stephan Van den Broucke, Patrick Bodenmann.

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
