## [Decision Letter · Decision Letter 0]

8 Jan 2023

PONE-D-22-20725How socioeconomically disadvantaged people access, understand, appraise, and apply health information: a qualitative study exploring health literacy skillsPLOS ONE

Dear Dr. Stormacq,

Thank you for submitting your manuscript to PLOS ONE. After careful consideration, we feel that it has merit but does not fully meet PLOS ONE’s publication criteria as it currently stands. Therefore, we invite you to submit a revised version of the manuscript that addresses the points raised during the review process.

We look forward to receiving your revised manuscript.

Kind regards,

Nabeel Al-Yateem, PhD

Academic Editor

PLOS ONE

and https://journals.plos.org/plosone/s/file?id=ba62/PLOSOne_formatting_sample_title_authors_affiliations.pdf.

Reviewers' comments:

Reviewer's Responses to Questions

**Comments to the Author**

1. Is the manuscript technically sound, and do the data support the conclusions?

Reviewer #1: Yes

Reviewer #2: Yes

2. Has the statistical analysis been performed appropriately and rigorously? 

Reviewer #1: N/A

Reviewer #2: N/A

3. Have the authors made all data underlying the findings in their manuscript fully available?

Reviewer #1: Yes

Reviewer #2: No

4. Is the manuscript presented in an intelligible fashion and written in standard English?

Reviewer #1: Yes

Reviewer #2: Yes

5. Review Comments to the Author

Reviewer #1: This is a well written paper addressing and important topic. I commend the authors on this simple and high quality piece of research. I have only minor suggestions for its improvent as set out below:

1. In the introduction it would be helpful to include some background information about the health care financing model in Switzerland so that the readers can interpret the results with a fuller understanding. Much of this information is included in the discussion but I felt it was needed earlier in the papers to help readers interpret this important aspect of the results.

2. The sample size even for a qualitative study is small. This should be included as a limitation.

3. The age range of the sample is rather narrow. Participants range between ages 44-60. This should be included as a limitation and there should be some discussion about how this might have narrowed the kinds of responses we saw particularly regarding use of the internet. For instance I would expect if more younger participants were included there may be different or additional information about accessing health information online. Similarly people in the 70s and 80s face obstacles that are skill related to accessing online information. Acknowledgement of this limitation is needed in the discussion.

4. On page 13 could gross monthly income also be presented in Euro and or USD to for international readers.

5. The paper overall is very long. I would suggest trying to reduce the results section and the discussion section by around 25%. One way to help achieve this would be to edit down the quotes to more succinct examples. For instance all quotes could be reduced to no more than 3 lines long. Table 4 could be put into an appendix or online supplement.

6. Typo on line 839 – ‘immediately is repeated in this sentence.

Reviewer #2: Thank you for the opportunity to review this manuscript that deals with a current and important topic.

Background- The framework that supports the concept of health literacy was clearly established, as well as the study rationality and design. It is suggested to review more recent investigations that address or develop interventions with socioeconomically disadvantaged persons.

Methods - in general are adequate, however the authors cite on page 9 ('202) the use of a French version 16 scale items that evaluate the experienced difficulties in assessing, understanding, appraising and applying health information and it is not described as these results were treated and for what purpose. They are presented in table 2 of the results and in `1018. I did not get what is a CHF 15 gift card ('210).The researchers cite that: data saturation was reached after 10 participants, but 12 people were interviewed. Justify.

Results- four main themes with three pathways regarding health information are reported.

The table on pages 15-18 is excellent, however the following table and the description of the results are extremely extensive.

Discussion- the results obtained can be summarized (according to table 2) and not described again, as well as presenting the results of the cited papers in a grouped way, as done between `853 and `860. Although participants address pharmacists and health professional skills are cited in the discussion, it is not clear whether the health system provides primary health care or access to other health professionals. It is also unclear whether the participants sought health information because they had chronic diseases (looking for treatment and prevention of complications). in some sentences health promotion is mentioned. Also, digital skills are not discussed considering the age of study participants as well years of study and participants' access to internet. The fact that most of the participants are women is only mentioned in ˜limitations and strengths˜. The lack of fluency in French is only mentioned as a limitation of the study, but it seems to be an important limitation for pathways 1 and 2. Pathway 3 is not seemed to have been discussed.

6. PLOS authors have the option to publish the peer review history of their article (what does this mean?). If published, this will include your full peer review and any attached files.

Reviewer #1: **Yes: **Kirsten McCaffery

Reviewer #2: No

---

## [Author Response · Author response to Decision Letter 0]

22 Feb 2023

Dear editor, 

We thank the reviewers for their interesting and useful comments. We improved the manuscript based on the reviewers’ comments. Specifically, we have amended the text of the manuscript as outlined below. 

Reviewer # 1: 

1. In the introduction it would be helpful to include some background information about the health care financing model in Switzerland so that the readers can interpret the results with a fuller understanding. Much of this information is included in the discussion but I felt it was needed earlier in the papers to help readers interpret this important aspect of the results.

We thank the reviewer for this pertinent comment. To take it into account, the description of the financing of the Swiss health care system has been moved to the introduction on pages 4 and 5.

2. The sample size even for a qualitative study is small. This should be included as a limitation. The age range of the sample is rather narrow. Participants range between ages 44-60. This should be included as a limitation and there should be some discussion about how this might have narrowed the kinds of responses we saw particularly regarding use of the internet. For instance I would expect if more younger participants were included there may be different or additional information about accessing health information online. Similarly people in the 70s and 80s face obstacles that are skill related to accessing online information. Acknowledgement of this limitation is needed in the discussion.

We thank the reviewer for pointing out these limitations. We acknowledge that our sample is relatively small due to well-known difficulties in accessing and engaging disadvantaged groups in research projects. This observation has been added to the limitations of our study on page 45. Despite this limitation, we believe that our twelve interviews yielded rich data related to the health literacy process of socioeconomically disadvantaged people. 

We have also added some sentences about the age range of our sample in the limitations section on page 46. 

3. On page 13 could gross monthly income also be presented in Euro and or USD to for international readers.

We thank the reviewer for this relevant comment. The exchange rates (swiss francs/euros and swiss francs/dollars) at the data collection time have been added as a footnote in Table 2 (pages 14 and 15).

4. The paper overall is very long. I would suggest trying to reduce the results section and the discussion section by around 25%. One way to help achieve this would be to edit down the quotes to more succinct examples. For instance all quotes could be reduced to no more than 3 lines long. Table 4 could be put into an appendix or online supplement.

We acknowledge that the manuscript is substantial. To take this comment into account, we have followed all the recommendations proposed by the reviewer. We have revised and reduced all quotes exceeding 3 lines. However, for some, we made the choice not to reduce them too much so as not to lose the richness of the data. Table 3 has been deleted and is now proposed in Supporting Information. As recommended, the discussion section was also reduced.

5. Typo on line 839 – ‘immediately is repeated in this sentence.

We thank the reviewer for this careful reading. We apologize for this typing error and have accordingly deleted one of the repeated words. 

Reviewer # 2: 

1. Background- The framework that supports the concept of health literacy was clearly established, as well as the study rationality and design. It is suggested to review more recent investigations that address or develop interventions with socioeconomically disadvantaged persons.

We thank the reviewer or this interesting comment and agree that addressing the state of the scientific literature on the development of LS interventions among socioeconomically disadvantaged people was an important point to add to the introduction. This has therefore been added to pages 5 and 6.

2. Methods - in general are adequate, however the authors cite on page 9 ('202) the use of a French version 16 scale items that evaluate the experienced difficulties in assessing, understanding, appraising and applying health information and it is not described as these results were treated and for what purpose. They are presented in table 2 of the results and in `1018. I did not get what is a CHF 15 gift card ('210). The researchers cite that: data saturation was reached after 10 participants, but 12 people were interviewed. Justify.

We thank the reviewer for these comments. 

The questionnaire HLS-EU-Q16 was initially used to describe the sample. To clarify this point in the method section, we added a sentence on page 10. Due to a small sample size, the results could not be contrasted according to the participants’ level of HL. This observation has been added in the limitations section on page 45. 

All participant received a gift card with an amount of 15 CHF (swiss francs), as a way of thanking them for participating in the study. As such, a clarification has been added on page 11. 

Regarding data saturation, this was indeed achieved after 10 interviews. We made the choice to conduct two additional interviews to ensure data saturation. This explanation has been added in the Results section on page 13.

3. Results- four main themes with three pathways regarding health information are reported. The table on pages 15-18 is excellent, however the following table and the description of the results are extremely extensive.

We thank the reviewer for this comment, in accordance with the reviewer # 1. To shorten the results section, we have reduced all quotes exceeding 3 lines. Table 3 has been deleted and is now proposed in Supporting Information.

4. Discussion- the results obtained can be summarized (according to table 2) and not described again, as well as presenting the results of the cited papers in a grouped way, as done between `853 and `860. 

To take this comment into account, we have summarized the descriptive characteristics of the sample on page 13. 

5. Although participants address pharmacists and health professional skills are cited in the discussion, it is not clear whether the health system provides primary health care or access to other health professionals. 

We thank the reviewer for this relevant comment. In Switzerland, primary health care is still underdeveloped. Doctors (mainly general practitioners) and pharmacists are often the first point of entry into the health care system. To take this comment into account, one sentence has been added into the discussion section on page 39.

6. It is also unclear whether the participants sought health information because they had chronic diseases (looking for treatment and prevention of complications). in some sentences health promotion is mentioned.

The twelve interviews yielded rich data related to the strategies used by the participants when trying to access, understand, appraise, and apply health information across the three domains of the health continuum, that is healthcare, disease prevention and health promotion. To clarify this point, the health topics addressed by the participants during the interview had been added in the Table 2. 

7. Also, digital skills are not discussed considering the age of study participants as well years of study and participants' access to internet. The fact that most of the participants are women is only mentioned in ˜limitations and strengths˜. The lack of fluency in French is only mentioned as a limitation of the study, but it seems to be an important limitation for pathways 1 and 2. Pathway 3 is not seemed to have been discussed.

We thank the reviewer for these comments. 

In the limitations section, on page 46, we have added some sentences on the link between participants' age and digital skills. One sentence has also been added on page 41 about the sociodemographic factors associated with poor digital health literacy skills.

The lack of fluency in French was mainly a limit to data collection. It is indeed possible to think that a lack of fluency in French could have had an impact on pathways 1 and 2. However, this was not mentioned by the participants as a barrier, either when consulting the doctor (Pathway 1) or when searching for health information on the internet (Pathway 2). On the internet, several participants mentioned searching for information in their mother tongue. The lack of fluency in French therefore only appears as a minor limitation. 

Although an important element, the Pathway 3 was originally not developed to limit the length of the discussion. However, to take this comment into account, we added two paragraphs on this subject into the discussion section on page 45. 

Also in response to reviewer # 2, interviews transcripts in French were deposited on the SwissUbase repository, to ensure availability of material (www.swissubase.ch). A doi has been created to access it (https://doi.org/10.48657/1eqm-cm37). Please note that this link is not yet active, the data being in editing by the SwissUbase repository. 

We thank you again for the work done on this manuscript and remain at your disposal for any further information.

The authors.

---

## [Decision Letter · Decision Letter 1]

2 May 2023

PONE-D-22-20725R1How socioeconomically disadvantaged people access, understand, appraise, and apply health information: a qualitative study exploring health literacy skillsPLOS ONE

Dear Dr. Stormacq,

Thank you for submitting your manuscript to PLOS ONE. After careful consideration, we feel that it has merit but does not fully meet PLOS ONE’s publication criteria as it currently stands. Therefore, we invite you to submit a revised version of the manuscript that addresses the points raised during the review process.

We look forward to receiving your revised manuscript.

Kind regards,

Nabeel Al-Yateem, PhD

Academic Editor

PLOS ONE

Reviewers' comments:

Reviewer's Responses to Questions

**Comments to the Author**

1. If the authors have adequately addressed your comments raised in a previous round of review and you feel that this manuscript is now acceptable for publication, you may indicate that here to bypass the “Comments to the Author” section, enter your conflict of interest statement in the “Confidential to Editor” section, and submit your "Accept" recommendation.

Reviewer #2: All comments have been addressed

Reviewer #3: All comments have been addressed

Reviewer #4: All comments have been addressed

2. Is the manuscript technically sound, and do the data support the conclusions?

Reviewer #2: Partly

Reviewer #3: Yes

Reviewer #4: Yes

3. Has the statistical analysis been performed appropriately and rigorously? 

Reviewer #2: N/A

Reviewer #3: N/A

Reviewer #4: N/A

4. Have the authors made all data underlying the findings in their manuscript fully available?

Reviewer #2: Yes

Reviewer #3: Yes

Reviewer #4: No

5. Is the manuscript presented in an intelligible fashion and written in standard English?

Reviewer #2: Yes

Reviewer #3: Yes

Reviewer #4: Yes

6. Review Comments to the Author

Reviewer #2: All comments have been adressed. The manuscript is still very long and you did not included any new reference from 2022 or 2023.

Reviewer #3: One comment to note - the data of this study was collected between April and June 2017. Given that this is now April 2023, this needs acknowledged as a limitation of this study.

Reviewer #4: I commend research team for undertaking this important study and for actively recruiting socially-disadvantaged community members; this is not an easy task and is too often overlooked! The authors provide a thorough qualitative manuscript and have responded adequately to the previous reviewers’ comments. In response to Reviewer 1 (Comment 2) , it may also be useful to add the age range of participants to the Abstract. In addition, the authors could do more to address Reviewer 2 (Comment 4), particularly by shortening the discussion such that the main findings are summarised rather than being described again.

In addition to this, I have some other comments for the authors to reflect on and address:

Typographical errors:

• Page 11: “All the transcriptions were check entirely for accuracy” should be “All the transcriptions were checked entirely for accuracy”.

• I find the wording of the first theme slightly awkward: “Getting informed for not worsening a precarious financial situation”. I encourage the authors to consider slight revisions for clarity.

Data collection

• Data was collected in 2017. Much has happened since then, particularly COVID which changed the way that many of us receive health information. I encourage the authors to reflect on how changes since then might have impacted the data collected, for example, in the Strengths and Limitations section of the manuscript.

Related literature:

• Muscat DM, Shepherd HL, Morony S, Smith SK, Dhillon HM, Trevena L, Hayen A, Luxford K, Nutbeam D, McCaffery K. Can adults with low literacy understand shared decision making questions? A qualitative investigation. Patient Educ Couns. 2016 Nov;99(11):1796-1802. doi: 10.1016/j.pec.2016.05.008. Epub 2016 May 9. PMID: 27344226.

• Jordan JE, Buchbinder R, Osborne RH. Conceptualising health literacy from the patient perspective. Patient Educ Couns. 2010 Apr;79(1):36-42. doi: 10.1016/j.pec.2009.10.001. Epub 2009 Nov 5. PMID: 19896320.

• Edwards M, Wood F, Davies M, Edwards A. 'Distributed health literacy': longitudinal qualitative analysis of the roles of health literacy mediators and social networks of people living with a long-term health condition. Health Expect. 2015 Oct;18(5):1180-93. doi: 10.1111/hex.12093. Epub 2013 Jun 17. PMID: 23773311; PMCID: PMC5060848.

• Muscat DM, Gessler D, Ayre J, Norgaard O, Heuck IR, Haar S, Maindal HT. Seeking a deeper understanding of 'distributed health literacy': A systematic review. Health Expect. 2022 Jun;25(3):856-868. doi: 10.1111/hex.13450. Epub 2022 Feb 18. PMID: 35178823; PMCID: PMC9122402.

7. PLOS authors have the option to publish the peer review history of their article (what does this mean?). If published, this will include your full peer review and any attached files.

Reviewer #2: **Yes: **Lisiane M G Paskulin

Reviewer #3: No

Reviewer #4: No

---

## [Author Response · Author response to Decision Letter 1]

15 Jun 2023

Dear editor, 

We thank the reviewers for their comments. We improved the manuscript based on the reviewers’ comments. Specifically, we have amended the text of the manuscript as outlined below. 

Reviewer # 2: 

1. The manuscript is still very long and you did not included any new reference from 2022 or 2023.

We thank the reviewer for this comment. To take it into account, we have reduced the results and the discussion sections. Regarding references, they have been updated, and references from 2022 and 2023 have been added. In addition, the references list has been reduced.

Reviewer # 3: 

1. One comment to note - the data of this study was collected between April and June 2017. Given that this is now April 2023, this needs acknowledged as a limitation of this study.

We thank the reviewer for pointing out this limitation. This observation has been added to the limitations of our study on page 41.

Reviewer # 4:

1. It may also be useful to add the age range of participants to the Abstract.

As asked, the age range of our participants has been added to the abstract.

2. The authors could do more to address Reviewer 2 (Comment 4), particularly by shortening the discussion such that the main findings are summarised rather than being described again.

We acknowledge that the manuscript was still very long. To take this comment into account, we have revised and reduced the discussion section as much as possible.

3. Typographical errors: Page 11: “All the transcriptions were check entirely for accuracy” should be “All the transcriptions were checked entirely for accuracy”.

We thank the reviewer for this careful reading. We apologize for this typing error and have accordingly corrected it.

4. I find the wording of the first theme slightly awkward: “Getting informed for not worsening a precarious financial situation”. I encourage the authors to consider slight revisions for clarity.

We thank the reviewer for this interesting comment. To take it into account, the first theme has been modified on page 17 for better clarity.

5. Data was collected in 2017. Much has happened since then, particularly COVID which changed the way that many of us receive health information. I encourage the authors to reflect on how changes since then might have impacted the data collected, for example, in the Strengths and Limitations section of the manuscript.

We thank the reviewer for pointing out this limitation. The impact of the COVID-19 pandemics on health literacy skills has been added to the limitations of our study on page 41.

We thank you again for the work done on this manuscript and remain at your disposal for any further information.

The authors.

---

## [Decision Letter · Decision Letter 2]

26 Jun 2023

How socioeconomically disadvantaged people access, understand, appraise, and apply health information: a qualitative study exploring health literacy skills

PONE-D-22-20725R2

Dear Dr. Stormacq,

We’re pleased to inform you that your manuscript has been judged scientifically suitable for publication and will be formally accepted for publication once it meets all outstanding technical requirements.

Kind regards,

Nabeel Al-Yateem, PhD

Academic Editor

PLOS ONE

Additional Editor Comments (optional):

Reviewers' comments:

Reviewer's Responses to Questions

**Comments to the Author**

1. If the authors have adequately addressed your comments raised in a previous round of review and you feel that this manuscript is now acceptable for publication, you may indicate that here to bypass the “Comments to the Author” section, enter your conflict of interest statement in the “Confidential to Editor” section, and submit your "Accept" recommendation.

Reviewer #3: All comments have been addressed

Reviewer #4: All comments have been addressed

2. Is the manuscript technically sound, and do the data support the conclusions?

Reviewer #3: Yes

Reviewer #4: Yes

3. Has the statistical analysis been performed appropriately and rigorously? 

Reviewer #3: N/A

Reviewer #4: N/A

4. Have the authors made all data underlying the findings in their manuscript fully available?

Reviewer #3: Yes

Reviewer #4: Yes

5. Is the manuscript presented in an intelligible fashion and written in standard English?

Reviewer #3: Yes

Reviewer #4: Yes

6. Review Comments to the Author

Reviewer #3: (No Response)

Reviewer #4: (No Response)

7. PLOS authors have the option to publish the peer review history of their article (what does this mean?). If published, this will include your full peer review and any attached files.

Reviewer #3: No

Reviewer #4: No

---

## [Editor Report · Acceptance letter]

19 Jul 2023

PONE-D-22-20725R2 

How socioeconomically disadvantaged people access, understand, appraise, and apply health information: a qualitative study exploring health literacy skills 

Dear Dr. Stormacq:

I'm pleased to inform you that your manuscript has been deemed suitable for publication in PLOS ONE. Congratulations! Your manuscript is now with our production department. 

Kind regards, 

on behalf of

Dr. Nabeel Al-Yateem 

Academic Editor

PLOS ONE